# Explainable AI (XAI) models applied to planning in financial markets

**Eric Benhamou** [1,2,3]**, Jean Jacques Ohana** [2]**, David Saltiel** [2] **, Beatrice Guez** [2]

[1]Miles, Lamsade, Dauphine University, [2]Ai For Alpha, [3]EB Ai Advisory

### Abstract

Regime changes planning in financial markets is well known to be hard to explain and interpret. Can an asset manager explain clearly the intuition of his regime changes prediction on equity market ? To answer this question, we consider a gradient boosting decision trees (GBDT) approach to plan regime changes on S&P 500 from a set of 150 technical, fundamental and macroeconomic features. We report an improved accuracy of GBDT over other machine learning (ML) methods on the S&P 500 futures prices. We show that retaining fewer and carefully selected features provides improvements across all ML approaches. Shapley values have recently been introduced from game theory to the field of ML. This approach allows a robust identification of the most important variables planning stock market crises, and of a local explanation of the crisis probability at each date, through a consistent features attribution. We apply this methodology to analyse in detail the March 2020 financial meltdown, for which the model offered a timely out of sample prediction. This analysis unveils in particular the contrarian predictive role of the tech equity sector before and after the crash.

## Introduction

From an external point of view, the asset management industry is a well-suited industry to apply machine learning (Benhamou et al. 2020b) as large amount of data are available thanks to the revolution of electronic trading and the methodical collection of data by asset managers or their acquisition from data providers. In addition, machine based decision can help reducing emotional bias and taking rational and systematic investment choices (Benhamou et al. 2020c). A recent major breakthrough has been the capacity to explain the outcome of the machine learning decisions and build an intuition on it.

Indeed, regime changes planning in financial markets (Benhamou et al. 2020a) is well known to be hard to forecast and explain. The planning of equity crashes, although particularly challenging due to their infrequent nature and the non-stationary features of financial markets, has been the focus of several important works in the past decades. For instance, (Sornette and Johansen 2001) have proposed a deterministic log-periodic model with finite-time explosion to represent

equity prices bubbles and crashes. In more recent works, (Chatzis et al. 2018) and (Samitas, Kampouris, and Kenourgios 2020) have introduced machine-learning approaches to the planning of global equity crises, emphasizing the importance of cross-market contagion effects. Our goal in this work is to use an AI model that is accurate and explainable to solve the multi-agent system environment of financial markets. Hence, we introduce a gradient boosting decision tree (GBDT) approach to predict large falls in the S& P500 equity index, using a large set of technical, fundamental and macroeconomic features as predictors. Besides illustrating the value of carefully selecting the features and the superior accuracy of GBDT over other ML approaches in some types of small/imbalanced data sets classification problems, our main contribution lies in the explanation of the model predictions at any date. Indeed, from a practitioner viewpoint, understanding why a model provides certain planning is at least as important as its accuracy. Although the complexity of AI models is often presented as a barrier to a practitioner understanding of their local planning, the use of SHAP (SHapley Additive exPlanation) values, introduced for the first time by (Lundberg and Lee 2017) in machine-learning applications, makes AI models more explainable and transparent. Shapley values are the contributions of each individual feature to the overall crash logit probability. They represent the only set of attributions presenting certain properties of consistency and additivity, as defined by Lundberg. In particular, the most commonly used variable importance measurement methodologies fail to pass the consistency test, which makes it difficult to compare their outputs across different models. Shapley values enlighten both the global understanding and the local explanations of machine learning prediction models, as they may be computed at each point in time.

In our context, this approach first allows us to determine which features most efficiently predict equity crashes (and in which global direction). We infer from a features importance analysis that the S&P500 crash probability is driven by a mix of pro-cyclical and counter-cyclical features. Pro-cyclical features consist either of positive economic/equity market developments that remove the prospect of large equity price drops, or alternatively of negative economic/equity market shocks that portend deadly equity downward spirals. Among these pro-cyclical features, we find the 120-day S&P500

Price/Earnings ratio percent change, the global risk aversion level, the 20-day S&P500 sales percent change, the six-months to one-year US 2 Yrs and 10 Yrs rates evolution, economic surprises indices, and the medium-term industrial metals, European equity indices and emerging currencies price trends. Conversely, counter-cyclical features can either be positive economic/equity market anticipations predating large equity price corrections, or negative shocks involving a reduced risk of equity downside moves. Important contrarian indicators are the put/call ratio, the six-months S&P500 sales percent change, the 100-day Nasdaq 100 Sharpe ratio and the U.S. 100-day 10 Yrs real interest change. A second crucial contribution of Shapley values is to help uncover how different features locally contribute to the logit probability at each point in time. We apply this methodology to analyse in detail the unfolding of the events surrounding the March 2020 equity meltdown, for which the model offered a timely prediction out of sample. On January 1, 2020, the crash probability was fairly low, standing at 9.4%. On February 3, we observe a first neat increase in the crash probability (to 27 %), driven by the 100-day Nasdaq Sharpe Ratio contrarian indicator. At the onset of the Covid crash, on March 2, 2020, most pro-cyclical indicators concurred to steeply increase the crash probability (to 61%) , as given by figure 5, in a way that proved prescient. Interestingly, the Nasdaq 100 index had already started its correction by this date, prompting the tech sector contrarian indicators to switch back in favor of a decreased crash probability. On April 1st, the crash probability plummeted back to 29 %, as the Nasdaq 100 appeared oversold while the Put/Call ratio reflected extremely cautious market anticipations. Overall, the analysis unveils the role of the tech sector as a powerful contrarian predictor before and after the March 2020 crash.

## Related works

Our work can be related to the ever growing field of machine learning applications to financial markets forecasting. Indeed, robust forecasting methods have recently garnered a lot of interest, both from finance scholars and practitioners. This interest can be traced back as early as the late 2000's where machine learning started to pick up. Instead of listing the large amount of works, we will refer readers to various works that reviewed the existing literature in chronological order.

In 2009, (Atsalakis and Valavanis 2009) surveyed already more than 100 related published articles using neural and neuro-fuzzy techniques derived and applied to forecasting stock markets, or discussing classifications of financial market data and forecasting methods. In 2010, (Li and Ma 2010) gave a survey on the application of artificial neural networks in forecasting financial market prices, including exchange rates, stock prices, and financial crisis prediction as well as option pricing. And the stream of machine learning was not only based on neural network but also genetic and evolutionary algorithms as reviewed in (Aguilar-Rivera, Valenzuela-Rendón, and Rodríguez-Ortiz 2015).

More recently, (Xing, Cambria, and Welsch 2018) reviewed the application of cutting-edge NLP techniques for financial forecasting, using text from financial news or tweets. (Rundo et al. 2019) covered the wider topic of machine learning, including deep learning, applications to financial portfolio allocation and optimization systems. (Nti, Adekoya, and Weyori 2019) focused on the use of support vector machine and artificial neural networks to forecast prices and regimes based on fundamental and technical analysis. Later on, (Shah, Isah, and Zulkernine 2019) discussed some of the challenges and research opportunities, including issues for algorithmic trading, back testing and live testing on single stocks and more generally prediction in financial market. Finally, (Sezer, Gudelek, and Ozbayoglu 2019) reviewed deep learning as well as other machine learning methods to forecast financial time series. As the hype has been recently mostly on deep learning, it comes as no surprise that most reviewed works relate to this field. One of the only works, to our knowledge, that refers to gradient boosted decision tree applications is (Krauss, Do, and Huck 2017)

Interestingly, Gradient boosting decision trees (GBDT) are almost non-existent in the financial market forecasting literature. As is well-knwon, GBDT are prone to over-fitting in regression applications. However, they are the method of choice for classification problems as reported by the ML platform Kaggle. In finance, the only space where GBDT have become popular is the credit scoring and retail banking literature. For instance, (Brown and Mues 2012) or (Marceau et al. 2019) reported that GBDT are the best ML method for this specific task as they can cope with limited amount of data and very imbalanced classes.

When classifying stock markets into two regimes (a 'normal' one and a 'crisis' one), we are precisely facing very imbalanced classes and a binary classification challenge. In addition, when working with daily observations, we are faced with a ML problem with a limited number of data. These two points can seriously hinder the performance of deep learning algorithms that are well known to be data greedy. Hence, our work investigates whether GBDT can provide a suitable method to identify stock market regimes. In addition, as a byproduct, GBDT provide explicit decision rules (as opposed to deep learning), making it an ideal candidate to investigate regime qualification for stock markets. In this work, we apply our methodology to the US S&P 500 futures prices. In unreported works, we have shown that our approach may easily and successfully be transposed to other leading stock indices like the Nasdaq, the Eurostoxx, the FTSE, the Nikkei or the MSCI Emerging futures prices.

Concerning explainable AI (XAI), there has been plenty of research on transparent and interpretable machine learning models, with comprehensive surveys like (Adadi and Berrada 2018) or (Choo and Liu 2018). (Rosenfeld and Richardson 2019) discusses at length XAI motivations, and methods. Furthermore, (Liu et al. 2017) regroup XAI into three main categories for understanding, diagnosing and refining. It also presents applicable examples relating to the prevailing state-of-the-art with upcoming future possibilities. Indeed, explainable systems for machine learning have has applied in multiple fields like plant stress phenotyping (Ghosal et al. 2018), heat recycling, fault detection (Madhikermi, Malhi, and Främling 2019), capsule Gastroenterology (Malhi et al. 2019-12) and loan attribtion (Malhi,

Knapic, and Framling 2020).Furthermore, (Malhi, Knapic, and Framling 2020) use a combination of LIME and Shapley for understanding and explaining models outputs. Our application is about financial markets which is a more complex multi-agent environment where global explainability is more needed, hence the usage of Shapley to provide intuition and insights to explain and understand model outputs.

## Contribution

Our contributions are threefold:

- We specify a valid GBDT methodology to identify stock market regimes, based on a combination of more than 150 features including financial, macro, risk aversion, price and technical indicators.

- We compare this methodology with other machine learning (ML) methods and report an improved accuracy of GBDT over other ML methods on the S& P 500 futures prices.

- Last but not least, we use Shapley values to provide a global understanding and local explanations of the model at each date, which allows us to analyze in detail the model predictions before and after the March 2020 equity crash.

## Why GBDT?

The motivations for Gradient boosting decision trees (GBDT) are multiple:

- GBDT are the most suitable ML methods for small data sets classification problems. In particular, they are known to perform better than their state-of-the-art cousins, Deep Learning methods, for small data sets. As a matter of fact, GBDT methods have been Kagglers' preferred ones and have won multiple challenges.

- GBDT methods are less sensitive to data re-scaling, compared to logistic regression or penalized methods.

- They can cope with imbalanced data sets.

- They allow for very fast training when using the leaf-wise tree growth (compared to level-wise tree growth).

## Methodology

In a normal regime, equity markets are rising as investors get rewarded for their risk-taking. This has been referred to as the 'equity risk premium' in the financial economics literature (Mehra and Prescott 1985). However, there are subsequent downturns when financial markets switch to panic mode and start falling sharply. Hence, we can simply assume that there are two equity market regimes:

- a *normal* regime where an asset manager should be positively exposed to benefit from the upward bias in equity markets.

- and a *crisis* regime, where an asset manager should either reduce its equity exposure or even possibly short-sell when permitted.

We define a crisis regime as an occurrence of index return below the historical $5\%$ percentile, computed on the training data set. The $5\%$ is not taken randomly but has been validated historically to provide meaningful levels, indicative of real panic and more importantly forecastable. For instance, in the S&P 500 market, typical levels are returns of -6 to -5 % over a 15-day horizon. To predict whether the coming 15-day return will be below the $5\%$ percentile (hence being classified as in crisis regime), we use more than 150 features described later on. Simply speaking, these 150 features are variables ranging from risk aversion measures to financial metrics indicators like 12-month-forward sales estimates, earning per share, Price/Earnings ratio, economic surprise indices (like the aggregated Citigroup index that compiles major figures like ISM numbers, non farm payrolls, unemployment rates, etc).

We only consider two regimes with a specific focus on left-tail events on the returns distribution because we found it easier to characterize extreme returns than to plan outright returns using our set of financial features. In the ML language, our regime detection problem is a pure supervised learning exercise, with a two-regimes classification. Hence the probability of being in the normal regime and the one of being in the crisis regime sum to one.

Daily price data are denoted by $P_t$. The return over a period of $d$ trading days is simply given by the corresponding percentage change over the period: $R_t^d = P_t/P_{t-d} - 1$. The crisis regime is determined by the subset of events where returns are lower or equal to the historical 5% percentile denoted by $C$. Returns that are below this threshold are labeled "1" while the label value for the normal regime is set to "0". Using traditional binary classification formalism, we denote the training data $X = \{x_i\}_{i=1}^N$ with $x_i \in \mathbb{R}^D$ and their corresponding labels $Y = \{y_i\}_{i=1}^N$ with $y_i \in 0, 1$. The goal is to find the best *classification* function $f^*(x)$ according to the temporal sum of some specific loss function $\mathcal{L}(y_i, f(x_i))$ as follows:

$$f^* = \arg\min_f \sum_{i=1}^N \mathcal{L}(y_i, f(x_i))$$

Gradient boosting assumes the function $f$ to take an additive form :

$$f(x) = \sum_{m=1}^T f_m(x) \tag{1}$$

where $T$ is the number of iterations. The set of weak learners $f_m(x)$ is designed in an incremental fashion. At the $m$-th stage, the newly added function, $f_m$ is chosen to optimize the aggregated loss while keeping the previously found weak learners $\{f_j\}_{j=1}^{m-1}$ fixed. Each function $f_m$ belongs to a set of parameterized base learners that are modeled as decision trees. Hence, in GBDT, there is an obvious design trade-off between taking a large number of boosted rounds and very simple based decision trees or a limited number of base learners but of larger size. From our experience, it is better to take small decision trees to avoid over-fitting and an important number of boosted rounds. In this work, we use $500$ boosted rounds. The intuition between this choice

is to prefer a large crowd of experts that difficultly memorize data and should hence avoid over-fitting compared to a small number of strong experts that are represented by large decision trees. Indeed, if these trees go wrong, their failure is not averaged out, as opposed to the first alternative. Typical implementations of GBDT are XGBoost, as presented in (Chen and Guestrin 2016), LightGBM as presented (Ke et al. 2017), or Catboost as presented (Prokhorenkova et al. 2018). We tested both XGBoost and LightGBM and found threefold speed for LighGBM compared to XGBoost for similar learning performances. Hence, in the rest of the paper, we will focus on LightGBM.

For our experiments, we use daily observations of the S&P 500 merged back-adjusted (rolled) futures prices using Homa internal market data. Our daily observations are from 01Jan2003 to 15Jan2021. We split our data into three subsets: a training sample from 01Jan2003 to 31Dec2018, a validation sample used to find best hyper-parameters from 01Jan2019 to 31Dec2019 and a test sample from 01Jan2020 to 15 Jan2021.

## GBDT hyperparamers

The GBDT model contains a high number of hyper-parameters to be specified. From our experience, the following hyper-parameters are very relevant for imbalanced data sets and need to be fine-tuned using evolutionary optimisations as presented in (Benhamou et al. 2019): min sum hessian in leaf, min gain to split, feature fraction, bagging fraction and lambda l2. The max depth parameter plays a central role in the use of GBDT. On the S&P 500 futures, we found that very small trees with a max depth of one performs better over time than larger trees. The 5 parameters mentioned above are determined as the best hyper parameters on the validation set.

## Features used

The model is fed by more than 150 features to derive a daily 'crash' probability. These data can be grouped into 6 families:

- **Risk aversion metrics** such as equities', currencies' or commodities' implied volatilities, credit spreads and VIC forward curves.

- **Price indicators** such as returns, sharpe ratio of major stock markets, distance from long term moving average and equity-bond correlation.

- **Financial metrics** such as sales growth or Price/Earnings ratios forecast 12 month forward.

- **Macroeconomic indicators** such as economic surprises indices by region and globally as given by Citigroup surprise index.

- **Technical indicators** such as market breath or put-call ratio.

- **Rates** such as 10 Yrs and 2 Yrs U.S. rates, or break-even inflation information.

## Process of features selection

Using all raw features would add too much noise in our model and would lead to biased decisions. We thus need to select or extract only the most meaningful features. As we can see in figure 1, we do so by removing the features in 2 steps:

- Based on gradient boosting trees, we rank the features by importance or contribution.

- We then pay attention to the severity of multicollinearity in an ordinary least squares regression analysis by computing the variance inflation factor (VIF) to remove colinear features. Considering a linear model $Y = \beta_0 + \beta_1 X_1 + \beta_2 X_2 + .. + \beta_n X_n + \epsilon$, the VIF is equal to $\frac{1}{1-R_j^2}$, $R_j^2$ being the multiple $R^2$ for the regression of $X_j$ on other covariates. The VIF reflects the presence of collinear factors that increase the variance in the coefficient estimates.

At the end of this 2-part process, we only keep 33% of the initial features.

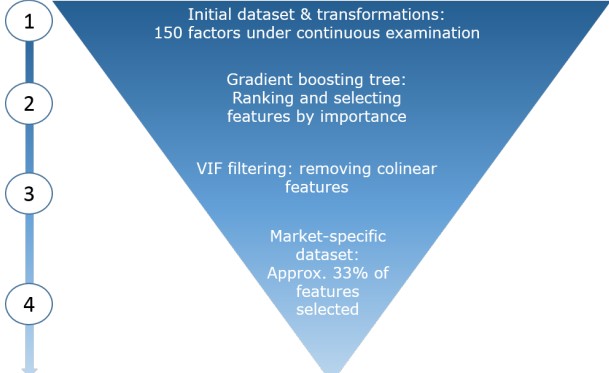

Figure 1: Probabilities of crash

In the next section, we will investigate whether removing correlated features improves the out-of-sample precision of the model.

## Results

### Model presentation

Although our work is mostly focused on the GBDT model, we compare it against common ML models, namely:

- a support vector model with a radial basis function kernel, a $\gamma$ parameter of 2 and a $C$ parameter of 1 (RBF SVM). We use the sklearn implementation. The two hyper parameters $\gamma$ and $C$ are found on the validation set.

- a Random Forest (RF) model, whose max depth is set to 1 and boosted rounds are set to 500. We purposely tune the RF model similarly to our GBDT model in order to benefit from the above mentioned error averaging feature. We found this parameter combination to perform well for annual validation data sets ranging from year 2015 onward on the S&P 500 market. We note that a max depth of 1 does not allow for interaction effects between features.

- a first deep learning model, referred to in our experiment as Deep FC (for fully connected layers), which is naively built with three fully connected layers (64, 32 and one for the final layer) with a drop out in of 5 % between and Relu activation, whose implementation details rely on tensorflow keras 2.0.

- a second more advanced deep learning model consisting of two layers referred to in our experiment as deep LSTM: a 64 nodes LSTM layer followed by a 5% dropout followed by a 32 nodes dense layer followed by a dense layer with a single node and a sigmoïd activation.

For both deep learning models, we use a standard Adam optimizer whose benefit is to combine adaptive gradient descent with root mean square propagation (Kingma and Ba 2014).

We train each model using either the full set of features or only the filtered ones, as described in 1. Hence, for each model, we add a suffix ' raw' or ' FS' to specify if the model is trained on the full set of features or after features selections. We provide the performance of these models according to different metrics, namely accuracy, precision, recall, f1-score, AUC and AUC-pr in tables 1 and 2. The GBDT with features selection is superior according to all metrics and outperforms in particular the deep learning model based on LSTM, confirming the consensus reached in the ML community as regards classification problems in small and imbalanced data sets.

Table 1: Model comparison

| Model | accuracy | precision | recall |
|---|---|---|---|
| GBDT FS | **0.89** | **0.55** | **0.55** |
| Deep LSTM FS | 0.87 | 0.06 | 0.02 |
| RBF SVM FS | 0.87 | 0.03 | 0.07 |
| Random Forest FS | 0.87 | 0.03 | 0.07 |
| Deep FC FS | 0.87 | 0.01 | 0.02 |
| Deep LSTM Raw | 0.84 | 0.37 | 0.33 |
| RBF SVM Raw | 0.87 | 0.02 | 0.01 |
| Random Forest Raw | 0.86 | 0.30 | 0.09 |
| GBDT Raw | 0.86 | 0.20 | 0.03 |
| Deep FC Raw | 0.85 | 0.07 | 0.05 |

Table 2: Model comparison

| Model | f1-score | auc | auc-pr |
|---|---|---|---|
| GBDT FS | **0.55** | **0.83** | **0.58** |
| Deep LSTM FS | 0.05 | 0.74 | 0.56 |
| RBF SVM FS | 0.06 | 0.50 | 0.56 |
| Random Forest FS | 0.04 | 0.54 | 0.56 |
| Deep FC FS | 0.04 | 0.50 | 0.56 |
| Deep LSTM Raw | 0.35 | 0.63 | 0.39 |
| RBF SVM Raw | 0.05 | 0.50 | 0.36 |
| Random Forest Raw | 0.14 | 0.53 | 0.25 |
| GBDT Raw | 0.05 | 0.51 | 0.18 |
| Deep FC Raw | 0.02 | 0.49 | 0.06 |

**AUC performance**

Figure 2 provides the ROC Curve for the two best performing models, the GBDT and the Deep learning LSTM model with features selection. ROC curves enables to visualize and analyse the relationship between precision and recall and to investigate whether the model makes more type I or type II errors when identifying market regimes. The receiver operating characteristic (ROC) curve plots the true positive rate (sensitivity) on the vertical axis against the false positive rate (1 - specificity, fall-out) on the horizontal axis for all possible threshold values. The two curves are well above the *blind guess* benchmark that is represented by the dotted red line. This effectively demonstrates that these two models have some predictability power, although being far from a perfect score that would be represented by a half square. Furthermore, the area under the GBDT curve with features selection is 0.83, to be compared with 0.74, the one of the second best model (deep LSTM), also with Features selection. Its curve, in blue, is mostly over the one of the second best model (deep LSTM), in red, which indicates that in most situations, GBDT model performs better than the deep LSTM model.

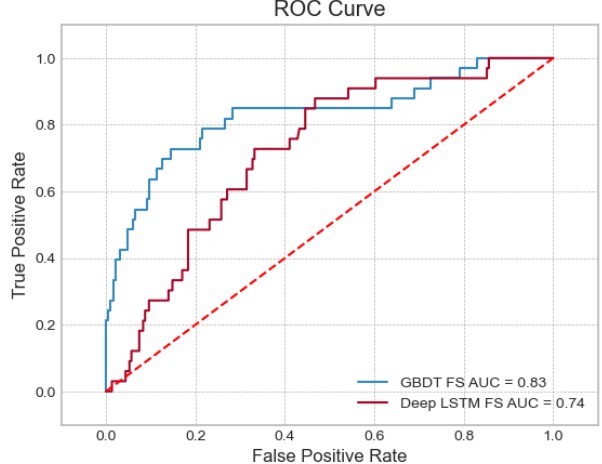

Figure 2: ROC Curve of the two best models

## Understanding the model

**Shapley values**

Building on the work of (Lundberg and Lee 2017), we use Shapley values to represent the contribution of each feature to the crisis probability. SHAP (SHapley Additive exPlanation) values explain the output of a function $f$ as a sum of the effects of each feature. It assigns an importance value to each feature that represents the effect on the model planning of including that feature. To compute this effect, a model $f_{S \cup \{i\}}$ is trained with that feature present, and another model $f_S$ is trained with the feature withheld. This method hence requires retraining the model on all feature subsets $S \subseteq M \setminus \{i\}$, where $M$ is the set of all features. Then, predictions from the two models are compared through the difference $f_{S \cup \{i\}}(x_{S \cup \{i\}})$ - $f_S(x_S)$, where $x_S$

represents the values of the input features in the set $S$. Since the effect of withholding a feature depends on other features in the model, the preceding differences are computed on all possible differences $f_{S\cup\{i\}}(x_{S\cup\{i\}})$ - $f_S(x_S)$, for all possible subsets $S \subseteq M\backslash\{i\}$. Shapley values are then constructed as a weighted average of all these differences, as follows: The Shapley value $\Phi_i$ attributed to feature $i$ is defined as:

$$\Phi_i = \sum_{S\subseteq M\backslash\{i\}} \frac{|S|!\,(|M|-|S|-1)!}{|M|!}\left(f_{S\cup\{i\}}(x_{S\cup\{i\}}) - f_S(x_S)\right)$$

where $|A|$ refers to the cardinal of the set $A$, $M$ is the complete set of features, $S$ is the subset of features used, and $x_S$ represents the values of the input features in the set $S$. Proofs from game theory shows that Shapley values are the only possible consistent approach such that the sum of the feature attributions is equal to the output of the function we are to explain. Compared to LIME as presented in (Ribeiro, Singh, and Guestrin 2016), Shap has the advantage of consistency, and focus at global interpretability versus local for LIME.

The exact computation of SHAP values is challenging. In practice, assuming features' independence, we approximate $f_S(x_S)$ by the Shapley 'sampling value', i.e. the conditional mean of the global model prediction $\hat{f}(X)$ (calibrated on the complete set of features), marginalizing over the values $x_C$ of features that are not included in set $S$:

$$f_S(x_S) = \mathbb{E}[\hat{f}(X)|x_S] \approx \int \hat{f}(x_S, x_C)p(x_C)dx_C$$

In our application, with max depth of 1, interaction effects between features are discarded, which allows to compute Shapley values trivially from equation (1).

## Shapley interpretation

We can rank the Shapley values by order of magnitude importance, defined as the average absolute Shapley value over the training set of the model: this is essentially the average impact on model output when a feature becomes "hidden" from the model. Furthermore, the correlation of a feature to its Shapley value provides insight into the effect of this feature on the probability of a stock market crash. Figure 3 represents the joint behavior of Shapley and features values to better grasp their non-linear dependencies.

Concerning figure 3, we observe that the most significant feature is the 250 days percent change in S&P 500 Price Earnings ratio (using the forward 1 Yr Earnings of the index as provided by Bloomberg). This reflects the presence of persistent cycles during which market participants' bullish anticipations regarding future earnings growths and market valuations translate into reduced downside risk for equity prices.

By the same token, a positive (resp. negative) 250-day change in the US 2 Yrs yield characterizes a regime of growth (resp. recession) in equities. A positive change in the Bloomberg Base Metals index is associated to a reduced crash probability. The same reasoning applies to the FX Emerging Basket, the S&P Sales evolution, the Euro Stoxx distance to its 200-day moving average and the EU Economic Surprise Index. Similarly, a higher (resp. lower)

Risk Aversion implies a higher (resp. lower) crash probability and the same relationship is observed for the realized 10-day S&P 500 volatility.

Interestingly, the model identifies the Put/Call ratio as a powerful contrarian indicator. Indeed, a persistently low level of the Put/Call ratio (as reflected by a low 20-day moving average) reflects overoptimistic expectations and therefore an under-hedged market. Last but not least, the Nasdaq 100 is identified as a contrarian indicator : the higher the Nasdaq 20-day percent change and the higher the Nasdaq 100-day and 250-day Sharpe Ratios, the higher the crash probability. More generally, foreign markets are used pro-cyclically (Euro Stoxx, BCOM Industrials, FX emerging) whereas most domestic price indicators are used counter-cyclically (Nasdaq 100, S&P 500). This is an example where we can see some strong added value from the machine learning approach over a human approach, as the former combines contrarian and trend following signals while the latter is generally biased towards one type of signals.

## Joint features and Shapley Values Distribution

Because some of the features have a strongly non-linear relation to the crisis probability, we also display in figure 3 the joint behavior of features and Shapley values at each point in time. The y-axis reports the Shapley values, i.e. the feature contributions to the model output in log-odds (we recall that the GDBT model has a logistic loss) while the color of the dot represents the value of that feature at each point in time. This representation uncovers the non-linearities in the relationship between the Shapley values and the features.

For instance, a large 250-day increase in the P/E ratio (in red color) has a negative impact on the crash probability, everything else equal. The same type of dependency is observed for the change in US 10 Yrs and 2 Yrs yields: the higher (resp. lower) the change in yield, the lower (resp. higher) the crash probability.

However, the dependency of the Shapley value to the 120-day BCOM Industrial Metals Sharpe ratio is non-linear. Elevated Sharpe ratios portend a lower crash probability while the relation vanishes for low Sharpe ratios. An ambiguous dependency is also observed for the 100-day Emerging FX Sharpe ratio, which explains its muted correlation to the Shapley value. This observation is all the more striking as this feature is identified as important in terms of its global absolute impact on the model output. By the same token, the 20-day percent change in S&P 500 Sales does not display a linear relationship to the crash probability. First of all, mostly elevated values of the change in sales are used by the model. Second, large increases in S&P 500 sales are most of the time associated with a drop in the crash probability, but not in every instance.

The impact of the distance of the Euro Stoxx 50 to its 200-day moving average is mostly unambiguous. Elevated levels in the feature's distribution generally (but not systematically) involve a decrease in the crash probability, whereas low levels of this feature portend an increased likelihood of crisis. The 20-day Moving Average of the Put/Call Ratio intervenes as a linear contrarian predictor of the crash probability.

The 20-day percent change in the Nasdaq 100 price is confirmed as a contrarian indicator. As illustrated in Figure 3, the impact of the 20-day Nasqaq 100 returns is non-linear, as negative returns may predict strongly reduced crash probabilities, while positive returns result in a more moderate increase in the crash probability. Conversely, the 20-day Euro Stoxx returns have a pro-cyclical linear impact on the crash probability, as confirmed by figure 3. As previously stated, the GBDT model uses non-US markets in a pro-cyclical way and U.S. markets in a contrarian manner.

## Local explanation of the Covid March 2020 meltdown

Not only can Shapley values provide a global interpretation of features' impacts, as described in section  and in figure 3, but they can also convey local explanations at every single date.

The figure 4 provides the Shapley values for the model on February, 2020. At this date, the model was still positive on the S&P 500 as the crash probability was fairly low, standing at 9.4%. The 6% 120-day increase in the P/E ratio, the low risk aversion level, reflecting ample liquidity conditions, and the positive EU Economic Surprise index all concurred to produce a low crash probability. However, the decline in the US LIBOR rate, which conveyed gloomy projections on the U.S. economy, and the elevated Put/Call ratio, reflecting excessive speculative behavior, both contributed positively to the crash probability. On February 3, we observe a first steep increase in the crash probability, driven by the 100-day Nasdaq Sharpe Ratio contrarian indicator. At the onset of the Covid crash, on March 2, 2020, the crash probability dramatically increased on the back of deteriorating industrial metals dynamics, falling Euro Stoxx and FTSE prices, negative EU economic surprises and decreasing S&P 500 P/E, which caused the model to identify a downturn in the equities' cycle. This prediction eventually proved prescient. Interestingly, the Nasdaq 100 index had already started its correction by this date, prompting the tech sector contrarian indicators to switch back in favor of a decreased crash probability.

We can do the same exercise for April 1, 2020. The crash probability plummeted as contrarian indicators started to balance pro-cyclical indicators : the Nasdaq 100 appeared oversold while the Put/Call ratio reflected extremely cautious market anticipations. During several months, the crash probability stabilized between 20% and 30% until the start of July, which showed a noticeable decline of the probability to 11.2%. The P/E cycle started improving and the momentum signals on base metals and other equities started switching side. Although the crash probability fluctuated, it remained contained throughout the rest of the year.

Last but not least, if we look at Shapley value at the beginning of December 2020, we can draw further conclusions. At the turn of the year, most signals were positive on the back of improving industrial metals, recovering European equity markets dynamics, and improving liquidity conditions (reflected by a falling dollar index and a low Risk Aversion). Although this positive picture is balanced by falling LIBOR rates and various small contributors, the features' vote sharply leans in favor of the bullish side.

## Conclusion

In this paper, we have shown how the GBDT method may classify financial markets into normal and crisis regimes, using 150 technical and fundamental features. When applied to the S&P 500, the method yields a high out-of-sample AUC score, which suggests that the machine is able to efficiently learn from previous crises. Our approach also displays an improved accuracy compared to other ML methods, confirming the relevance of GBDT in solving highly imbalanced classification problems with a limited number of observations. AI models complexity are often a barrier to a practitioner understanding of their local predictions. Yet, from the practitioner viewpoint, understanding why a model provides a certain prediction as at least as important as the accuracy of this prediction. Shapley values allow for a global understanding of the model behavior and for a local explanation of each feature's contribution to the crash probability at each observation date. This framework shed light on the unfolding of the model predictions during the events that surrounded the March 2020 equity meltdown. In particular, we unveiled the role of the tech equity sector as a powerful contrarian predictor during this episode.

A few caveats are in order to conclude this paper. First, the model is short-term in nature and should be employed with an agile flexible mindset, rather than to guide strategic investment decisions. Second, one must be careful not becoming overconfident about the forecasting ability of the model, even on a short-term horizon as the model only captures a *probability* of crash risk, generating false positive and false negative signals. More importantly, as financial markets exhibit a strongly non stationary behavior, it is subject to large out of sample prediction errors should new patterns emerge.

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

Figure 4: Shapley values for 2020-01-01

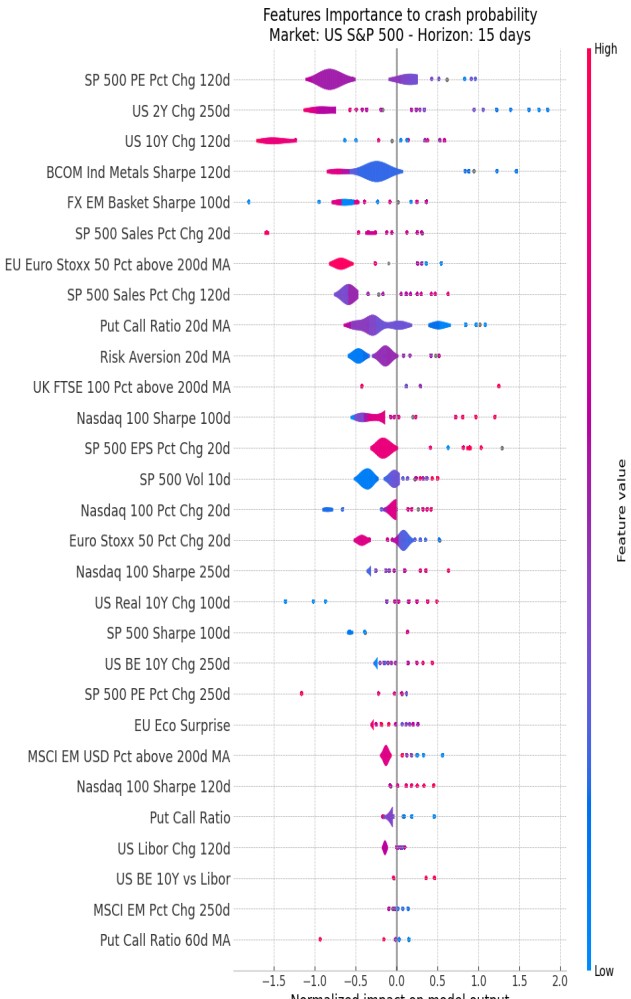

Figure 3: Marginal contribution of features with full distribution

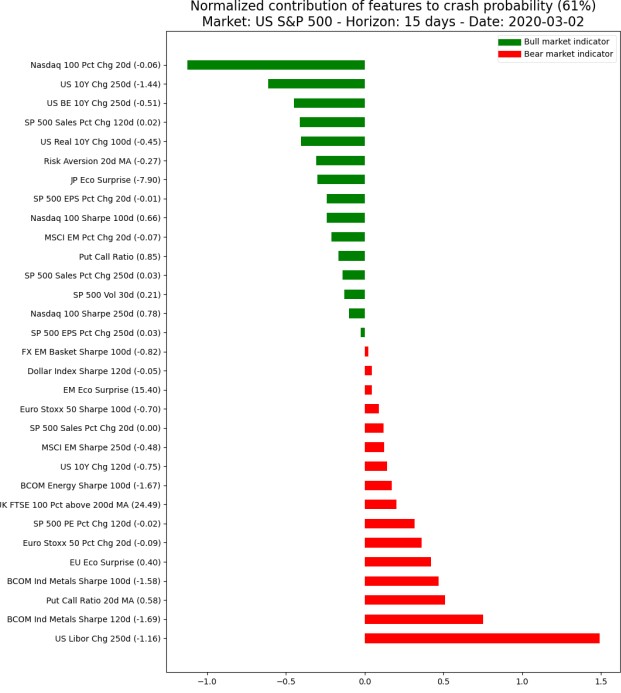

Figure 5: Shapley values for 2020-03-02

Benhamou, E.; Saltiel, D.; Vérel, S.; and Teytaud, F. 2019. BCMA-ES: A Bayesian approach to CMA-ES. *CoRR* abs/1904.01401.

Brown, I.; and Mues, C. 2012. An experimental comparison of classification algorithms for imbalanced credit scoring data sets. *Expert Systems with Applications* 39(3): 3446–3453. ISSN 0957-4174.

Chatzis, S.; Siakoulis, A. P. V.; Stavroulakis, E.; and Vlachogiannakis, N. 2018. Forecasting stock market crisis events using deep and statistical machine learning techniques. *Expert Systems with Applications* 112: 353–371.

Chen, T.; and Guestrin, C. 2016. XGBoost: A Scalable Tree Boosting System. *CoRR* abs/1603.02754.

Choo, J.; and Liu, S. 2018. Visual Analytics for Explainable Deep Learning. *CoRR* abs/1804.02527.

Ghosal, S.; Blystone, D.; Singh, A. K.; Ganapathysubramanian, B.; Singh, A.; and Sarkar, S. 2018. An explainable deep machine vision framework for plant stress phenotyping. *Proceedings of the National Academy of Sciences* 115(18): 4613–4618.

Ke, G.; Meng, Q.; Finley, T.; Wang, T.; Chen, W.; Ma, W.; Ye, Q.; and Liu, T.-Y. 2017. LightGBM: A Highly Efficient Gradient Boosting Decision Tree. In Guyon, I.; Luxburg, U. V.; Bengio, S.; Wallach, H.; Fergus, R.; Vishwanathan, S.; and Garnett, R., eds., *Advances in Neural Information Processing Systems*, volume 30, 3146–3154. Curran Associates, Inc.

Kingma, D.; and Ba, J. 2014. Adam: A Method for Stochastic Optimization.

Krauss, C.; Do, X. A.; and Huck, N. 2017. Deep neural networks, gradient-boosted trees, random forests: Statistical arbitrage on the S&P 500. *European Journal of Operational Research* 259(2): 689–702.

Li, Y.; and Ma, W. 2010. Applications of Artificial Neural Networks in Financial Economics: A Survey. In *2010 International Symposium on Computational Intelligence and Design*, volume 1, 211–214.

Liu, S.; Wang, X.; Liu, M.; and Zhu, J. 2017. Towards Better Analysis of Machine Learning Models: A Visual Analytics Perspective. *CoRR* abs/1702.01226.

Lundberg, S.; and Lee, S.-I. 2017. A Unified Approach to Interpreting Model Predictions.

Madhikermi, M.; Malhi, A.; and Främling, K. 2019. Explainable Artificial Intelligence Based Heat Recycler Fault Detection in Air Handling Unit. In *EXTRAAMAS@AAMAS*.

Malhi, A.; Kampik, T.; Pannu, H. S.; Madhikermi, M.; and Främling, K. 2019-12. Explaining Machine Learning-based Classifications of in-vivo Gastral Images. In *2019 Digital Image Computing: Techniques and Applications (DICTA)*, 7.

Malhi, A.; Knapic, S.; and Framling, K. 2020. Explainable Agents for Less Bias in Human-Agent Decision Making. In *EXTRAAMAS@AAMAS*. Springer.

Marceau, L.; Qiu, L.; Vandewiele, N.; and Charton, E. 2019. A comparison of Deep Learning performances with others machine learning algorithms on credit scoring unbalanced data. *CoRR* abs/1907.12363.

Mehra, R.; and Prescott, E. 1985. The equity premium: A puzzle. *Journal of Monetary Economics* 15(2): 145–161.

Nti, I. K.; Adekoya, A. F.; and Weyori, B. A. 2019. A systematic review of fundamental and technical analysis of stock market predictions. *Artificial Intelligence Review* 1–51.

Prokhorenkova, L.; Gusev, G.; Vorobev, A.; Dorogush, A. V.; and Gulin, A. 2018. CatBoost: unbiased boosting with categorical features. In Bengio, S.; Wallach, H.; Larochelle, H.; Grauman, K.; Cesa-Bianchi, N.; and Garnett, R., eds., *Advances in Neural Information Processing Systems*, volume 31, 6638–6648. Curran Associates, Inc.

Ribeiro, M. T.; Singh, S.; and Guestrin, C. 2016. "Why Should I Trust You?": Explaining the Predictions of Any Classifier. In *Proceedings of the 22nd ACM SIGKDD International Conference on Knowledge Discovery and Data Mining*, 1135–1144. Association for Computing Machinery.

Rosenfeld, A.; and Richardson, A. 2019. Explainability in Human-Agent Systems. *CoRR* abs/1904.08123.

Rundo, F.; Trenta, F.; di Stallo, A. L.; and Battiato, S. 2019. Machine Learning for Quantitative Finance Applications: A Survey. *Applied Sciences* 9(24): 5574.

Samitas, A.; Kampouris, E.; and Kenourgios, D. 2020. Forecasting stock market crisis events using deep and statistical machine learning techniques. *International Review of Financial Analysis* 71: 101507.

Sezer, O. B.; Gudelek, M. U.; and Ozbayoglu, A. M. 2019. Financial Time Series Forecasting with Deep Learning: A Systematic Literature Review: 2005-2019. *arXiv preprint arXiv:1911.13288* .

Shah, D.; Isah, H.; and Zulkernine, F. 2019. Stock Market Analysis: A Review and Taxonomy of Prediction Techniques. *International Journal of Financial Studies* 7(2): 26.

Sornette, D.; and Johansen, A. 2001. Significance of log-periodic precursors to financial crashes. *Quantitative Finance* 1: 452–471.

Xing, F. Z.; Cambria, E.; and Welsch, R. E. 2018. Natural language based financial forecasting: a survey. *Artificial Intelligence Review* 50(1): 49–73.
