# OpenReview forum: "Explainable AI (XAI) models applied to planning in financial markets"
_icaps-conference.org/ICAPS/2021/Workshop/XAIP — Reject_

### Official Review · AnonReviewer1 · 2021-07-04
**Does not fit well to the XAIP workshop**

**Rating:** 2
**Confidence:** 5

**Review:**

The paper at hand presents a case study where decision trees are applied to a finance data set to predict stock market crash probabilities. Accuracy of the resulting decision tree is compared with other AI model types. Finally, SHAP is applied to gain understanding of the learnt model. Although all this is well explained, it clearly is not within the scope of the workshop. Most significantly, it makes no use of AI planning. The term "planning" employed in this paper has a different meaning, pointing at some underlying misunderstanding maybe. The paper will fit better to some audience with special interest in data science for the financial domain.

---

### Official Review · AnonReviewer2 · 2021-07-05
**Seems sophisticated with respect to forecasting in finance, but is about a classification problem that has no aspects of sequential decision making or notion of planning.**

**Rating:** 3
**Confidence:** 3

**Review:**


Summary:
The authors present an approach to using Gradient Boosted Decision Trees to predict regime changes for equity markets using S&P 500 futures prices. The work includes feature selection to reduce the multicollinearity problem using variance inflation factor. The feature importance for the predictions are explained with Shapley values.  They analyze the shapley values for different features, and determine what type of indicators the features are. Lastly, they show how their method performs in the context of the financial crash after covid.


Review
---------------------------------
Related Work section:
The paper mentions the work by Krauss, Do, and Huck 2017, since they also considered GDBT for financial applications, but no more is said. How does your work compare to this ?
With respect to the statement “For instance, (Brown and Mues 2012) or (Marceau et al. 2019) reported that GBDT are the best ML method for this specific task as they can cope with limited amount of data and very imbalanced classes”. I looked into the papers. Marceau 2019 is a limited analysis of methods and they only conclude that both GBDT and Random Forests are better than SVM and Deep NN for their setup. I agree the numbers tilt a little in GBDT’s favor. However, Brown and Mues (2012) work which seems to have done a more extensive comparison of methods on unbalanced credit score dataset doesn’t consistently show GBDT as being the best ranking method; see Figures 2,3,4, and 5 in that paper. In fact Random Forests are consistently better than GBDT. So I think it is too strong a statement to say GBDT is the best ML method for imbalanced data sets. Also, in the setting where the data given for training was the least (70/30 split), Least Square SVMs performed the best. So I don’t think the papers support the statement that GBDT is the best for either imbalanced data and/or limited data in credit scoring. GBDT certainly seems like a good choice given the results in those papers. If there are other attractive reasons for GBDT like those mentioned in the section “Why GBDT”, then highlight those instead. Also some of the statements made in that section titled “why GBDT” needs citations; for example, which papers analyzed and concluded that “GBDT methods are less sensitive to data re-scaling, compared to logistic regression or penalized methods.”
------------------------
Methodology section:
	In the 4th para, the definition x_i \in R^D; you previously used “d” to refer to the number of training days, so use the lowercase “d” unless it is another variable. If it is another variable, please define it.

GBDT hyperparameters:
	It is written that “min sum hessian in leaf, min gain to split, feature fraction, bagging frac-tion and lambda l2” are the important hyperparameters, and need to be fine tuned using evolutionary optimization. The actual hyperparameters used in the experiments is not given ? This is important for any hope of reproducing any results. On that note, would your dataset of S&P 500 futures prices be collected and made publicly available for other researchers to evaluate ? If others are expected to manually collect the same data again using the dates given in the paper in order to verify your results, there is the potential for data mismatch making reproducibility harder.

Shapley Values:
please cite the source for the approximation method and formula used to calculate the approximate SHAP values; this is so we know if this is a previously established approximation (which I think it is), or your own version.
------------------------
Overall

While the work is sophisticated with respect to forecasting in finance, it has no element of sequential decision making ; this work is about a classification problem. There is no concept of planning (decisions/actions over multiple time steps), and as such it is difficult to justify it for a workshop related to planning research.

The feature analysis is clearly a more sophisticated and financially-savvy analysis than I am capable of properly evaluating and appreciating. I am unsure if any of our audience would have the necessary know-how to appreciate the insights in the paper. Concepts like “contrarian indicator” seem to be assumed as common knowledge. This maybe true in a finance-focused conference or workshop, and not likely outside of it. This extends to other features and concepts like “equity bond correlation” mentioned without definitions. Perhaps the authors might consider the KDD workshop on machine learning in finance.
http://www.wikicfp.com/cfp/servlet/event.showcfp?eventid=131154&copyownerid=82400

---

### Meta-Review · Area_Chairs · 2021-07-07

**Recommendation:** Reject
**Confidence:** 4

**Metareview:**

Thank you again for submitting the paper, unfortunately, both reviewers agree that the paper doesn't seem to be a good fit for the XAIP workshop. As AnonReviewer1 pointed out this could be due to a misunderstanding of the term planning. The workshop is generally focused on the problem of explanations for sequential decision-making settings and the paper doesn't seem to fit well into that category. But we hope the authors find the reviews still helpful, and AnonReviewer2 has also suggested another workshop that may be a better fit for the paper. To that list, I would also like to add the XAI workshops that are held during AAAI and IJCAI. These workshops look at a more diverse set of problems and I would imagine the paper would be a good fit for those workshops.

---

### Decision · Program_Chairs · 2021-07-08

Reject